# Analysis of the Results Determining the Positions and Velocities of Satellite Laser Ranging Stations during Earthquakes in 2010–2011

**Stanisław Schillak** [1,*] , **Agnieszka Satarowska** [2], **Dominik Sankowski** [2] **and Piotr Michałek** [1]

1 Centrum Badań Kosmicznych Polskiej Akademii Nauk (CBK PAN), Borowiec Astrogeodynamic Observatory, Bartycka 18A, 00-716 Warszawa, Poland; pmichalek@cbk.poznan.pl

2 Faculty of Electrical, Electronic, Computer and Control Engineering, Łódź University of Technology, ul. Bohdana Stefanowkiego 18, 90-924 Łódź, Poland; agnieszka.satarowska@dokt.p.lodz.pl (A.S.); dominik.sankowski@p.lodz.pl (D.S.)

* Correspondence: sch@cbk.poznan.pl; Tel.: +48-783-479-747

**Abstract:** This paper presents an analysis of the results determining the positions and the velocities of the 21 selected satellite laser ranging stations which performed observations from January 2008 to December 2012. This particularly interesting period of five years was selected, during which two strong earthquakes occurred near the stations. The focus was directed on the stations where the effects of the earthquakes were observed, i.e., Koganei (7308), Simosato (7838), and Changchun (7237) as a result of the tsunami in Japan on 11 February 2011, as well as Concepcion (7405IR) and San Juan (7406) as a result of the earthquake in Chile on 27 February 2010. The station positions were computed using the GSFC NASA GEODYN-II orbital software from the observations results of the satellites LAGEOS-1 and LAGEOS-2. The geocentric coordinates of the stations were determined from the normal equations of both satellites. The station velocities were computed from the positions determined for the observation epoch using the linear regression method. For each station, the following parameters were determined: mean total coordinates stability, standard deviation of determined positions and mean deviation from ITRF for topocentric components. For the best stations, the stability ranged from 4.0 mm to 6.5 mm, with the standard deviation of determined positions ranging from 0.9 mm to 1.7 mm. For two stations, a quadratic change in the station position was detected instead of a normal linear one after strong earthquakes.

**Keywords:** satellite geodesy; satellite laser ranging; satellite orbits; earthquakes; station position; station velocity

## 1. Introduction

One of the basic tasks of satellite geodesy is to determine the position and velocity of observation stations in a continuous mode [1]. The results of these measurements enable, above all, the creation of the International Terrestrial Reference Frame (ITRF), which is produced every few years (the latest one is ITRF2020) covering an increasing amount of measurement data. Determination of both the position and the velocity of the observation stations have been carried out continuously for many years based on four space techniques: satellite laser ranging (SLR), very long baseline interferometry (VLBI), global navigation satellite systems (GNSSs), and Doppler orbitography and radiopositioning integrated by satellite (DORIS) [2]. The data obtained in the frame of the measurements carried out using the above-mentioned techniques are the basis of the International Terrestrial Reference Frame (ITRF).

Satellite laser ranging represented by the International Laser Ranging Service (ILRS) [3,4] as an absolute measurement technique, on the basis of the obtained data, determine the position of the axis of the Earth's geocentric reference system XYZ and the center of

the system, which is the Earth's center of mass. Then, the Earth's rotation parameters are determined from the VLBI measurement data, and finally the coordinate system is significantly amplified by the position results obtained from the GNSS and DORIS satellites. Thus, the role of SLR measurements in this process is very important. The SLR technique is relatively old (the first measurements took place on 31 October 1964 on the satellite Beacon Explorer B [5]). Determining the coordinates of individual points on the Earth's surface and tracking changes in their position over time is a rich source of information on horizontal movements (N-S and E-W) caused mainly by the movement of tectonic plates. The vertical changes (U) results mainly from systematic errors of the SLR stations.

The aim of this work was to detect jumps and changes in the position and velocity of the SLR stations, and not to eliminate them. These effects are sources of very important information for SLR stations and SLR analysis data centers. For this purpose, the period 2008–2012, interesting in the authors' opinion, was selected. During this time, there were two strong earthquakes, which additionally caused three stations to move several centimeters. On the other hand, analysis centers, especially the ILRS Analysis Standing Committee have a different task. They have to determine the station coordinates as accurately as possible. It is especially important for determining the International Terrestrial Reference Frame, or satellites coordinates, where jumps should not occur.

The implementation of this work includes tracking the changes in the position of the SLR stations in the period under study, caused by natural factors such as earthquakes, as well as systematic and random errors of the stations. This article studies a particularly interesting period in the long history of satellite laser observations, covering the years 2010 and 2011, when several significant earthquakes occurred in the areas of operation of SLR stations: Concepcion (Chile), Koganei (Japan) and Monument Peak (California). Changes in the stations' position caused by these quakes were also observed at other nearby laser stations: San Juan (Argentina), Simosato (Japan), Changchun (China), Shanghai (China) and Komsomolsk-on-Amur (Russia). These significant changes in the station position and velocity resulted in the closure of several of these stations (Concepcion, San Juan, and Koganei). In order to check the results of the SLR stations operating in this period, the examined period was extended to 5 years, from 2008.0 to 2013.0, so that the velocity of the station could also be determined, which requires 3–5 years of observations to ensure adequate quality.

An important goal of this work is to determine the velocity of the stations; the knowledge of changes in position after very strong quakes is also especially important. Another aim of this work is also to assess the quality of individual stations, both from the point of view of systematic and random errors. Many SLR data analysis centers control the quality and quantity of the SLR observations, usually for a limited number of stations covering only selected parameters. Some examples of the most important SLR data analysis centers are ILRS ASC Product and Information Server [6], ILRS Monthly/Quarterly Global Performance Report Card [7], Multi-Satellite Bias Analysis Report [8], Combined Range Bias Report [9], and DGFI-TUM Analysis Center [10]. However, these centers do not provide a quick and detailed analysis of the results obtained by all the currently operating SLR stations.

## 2. Materials and Methods

### 2.1. Determining the Distance to Satellites

The task of a SLR station is to obtain precise measurements of the distance to the satellite and the epochs of these measurements. In addition, we need to know the values of pressure, temperature and humidity at the level of the station reference point to determine the tropospheric correction and the calibration correction from the measurements to the ground target (conducted before and after the observation, during the observation, or about every 1 h), which eliminate basic systematic errors and link the measurements to the station's reference point.

The result of the SLR observations has two values:

$\Delta t$—double distance to the satellite in picoseconds;

$T$—epoch of the laser shot in seconds with a UTC accuracy of 0.1 μs.

After filtering these results using the 3σ criterion, we calculate the observed distances between the $R_i$ station (reference point) and the satellite (center of mass).

$$R_i = \frac{(\Delta t_i - \Delta k)C}{2} + 2\Delta a_i + \Delta s \tag{1}$$

where:

$\Delta k$—calibration correction;

$\Delta a$—correction for reducing the speed of light when passing through the atmosphere (p, t, h, l, H—pressure, temperature, humidity, longitude, and geodetic height, respectively);

$\Delta s$—correction for the satellite center of mass (COM);

$C$—speed of light in vacuum = 299,792,458 m/s.

The obtained results are converted into normal points [11], which average in the range measurements in 120 s windows (for LAGEOS-1 and LAGEOS-2 satellites), and sent in the appropriate format to one of the two data centers (EUROLAS Data Center (EDC) or NASA Crustal Dynamics Data Information System (CDDIS)).

For the purposes of the present work, the results of the LAGEOS-1 and LAGEOS-2 satellites from the period 2008.0—2013.0 were downloaded from the EDC for all the laser stations operating in this period. Both LAGEOS satellites were selected due to their particular suitability for determining the station coordinates: long distance to satellites (approx. 6000 km), lack of atmospheric drag, small number of Earth's gravitational field coefficients (20 × 20), spherical shape ensuring a constant correction for the center of mass, and large amount of observations.

For the obtained results of all the stations in monthly periods, which were in the form of normal points of the individual passes, a sorting procedure was carried out according to the time of passes and conversion to the format required by the GSFC NASA GEODYN-II orbital software [12], which is the basis for computing the SLR stations coordinates (in total normal points for 42 stations). The program includes selected models and parameters necessary to perform computations of the orbital arcs, which are presented in Table 1.

**Table 1.** GEODYN II—force model and program parameters [13].

| Force models | |
| --- | --- |
| Earth gravity field | EGM2008 20 × 20 [14] |
| Earth tides | Convention IERS 2003 [15] |
| Ocean tides | GOT99.2 [16] |
| Third body gravity: moon, sun, and planets | DE403 [17] |
| Solar radiation pressure | Coefficient CR = 1.13 |
| Tide amplitudes—k2, k3, phase k2 | k2 = 0.3019, k3 = 0.093, phase k2 = 0.0 [18] |
| Earth albedo | [12] |
| Dynamic polar motion | [12] |
| Relativistic corrections | [12] |
| **Constants** | |
| Earth gravity parameter (GM) | $3.986004415 \times 10^{14} \ \mathrm{m^3/s^2}$ |
| Light velocity | 29,792.458 km/s |
| Semimajor axis of the Earth | 6378.13630 km |
| Inverse of the Earth's flattening | 298.25642 |

**Table 1.** *Cont.*

| Reference frame | |
|---|---|
| Inertial reference frame | J2000.0 |
| Coordinates reference system | true of date at 0 h of the first day of the each month |
| Stations coordinates | ITRF2008 [2] |
| Precession and nutation | IAU 2000 |
| Polar motion | C04 IERS |
| Tidal uplift | Love model h2 = 0.6078, l2 = 0.0847 [18] |
| Pole tide | [12] |
| **Estimated parameters** | |
| Satellite state vector | 6 parameters |
| Station geocentric coordinates | 3 parameters |
| Acceleration parameters | along-track, cross-track and radial at 5-days intervals |
| **Measurement model** | |
| Observations | 120 sec window of normal point, data from EUROLAS Data Center |
| Satellites | LAGEOS–1 and LAGEOS–2 |
| Center of mass correction | 25.1 cm |
| Cross—section area | 0.2827 m$^2$ |
| Mass | 406.965 kg (LAGEOS-1), 405.380 kg (LAGOES-2) |
| Laser pulse wavelength | 532 nm, 864 nm (7405IR) |
| Tropospheric refraction | Model Mendes–Pavlis [19,20] |
| **Editing criteria** | |
| Normal points residual | 5σ per arc |
| Cut-off | elevation 10o |
| Station coordinates cut-off | <50 normal points per station per arc |
| **Numerical integration** | |
| Integration | Cowell method |
| Orbit integration step size | 120 s |
| Arc length | 1 month |

### 2.2. Position of the SLR Station

Determination of the position and velocity of the observation stations differ significantly. The station position is computed via GEODYN-II software from the laser range measurements for monthly arc for a common reference epoch (2005.0) (detrended data). The velocities, on the other hand, are computed as the slope of the straight line of the coordinate components (XYZ or NEU) computed for the whole series of observation epoch (ITRF velocities are not needed then) (trended data).

The coordinates of the satellite are determined from the solution of the equation for the motion of the satellite.

$$\ddot{\mathbf{r}} = -\frac{\mu \mathbf{r}}{r^3} + \mathbf{A} \qquad (2)$$

where:

$\mathbf{r}$—geocentric satellite vector ($x, y, z, \dot{x}, \dot{y}, \dot{z}$);

$r$—the distance between the center of mass of the Earth (origin of the system) and the center of mass of the satellite;

$\mu$ = GM = 3.986004415 × 1014 m$^3$/s$^2$;

$\mathbf{A}$—the vector of the sum of accelerations from mechanical and gravitational forces.

As a result of the integration, we obtain the computed satellite coordinates, and by subtracting "a priori" the station positions from the ITRF ($X$, $Y$, $Z$) we have the distance to the satellite ($\rho$), which, if subtracted from the value of the observed distance $R_i$ (3), gives the difference of the observed-computed values (O–C) ($\Delta\rho$).

$$\rho_i = \sqrt{(X - x_i)^2 + (Y - y_i)^2 + (Z - z_i)^2}$$
$$\Delta\rho_i = (R_i - \rho_i) =\gg \min \tag{3}$$

We usually obtain about 10,000 of such observation equations per month for both LAGEOS satellites and using the Bayesian least squares method and subsequent iterations (usually 4 or 5), we determine the corrections to the $dP$ parameters using Formula (4).

$$O_i - C_i = -\sum_j \frac{\delta C_i}{\delta P_j} dP_j + dO_i \tag{4}$$

where:

$i$—$i$-th observation of $n$ ones;

$j$—$j$-th parameter of $m$ ones;

$dP_j$—correction to the $j$-th parameter;

$dO_i$—error in the $i$-th observation;

$\frac{\delta C}{\delta P}$—partial derivatives of the computed distance with respect to the parameter.

The basic parameters that we must determine are the satellite coordinates (6 parameters). Based on the choice of the other parameters depending on the tasks, we can determine, e.g., geocentric coordinates of the station (3 parameters), empirical acceleration coefficients, atmospheric drag coefficient, light pressure coefficient, pole coordinates, length of the day, coefficients of the Earth's gravitational field, tidal parameters, and physical constants.

### 2.3. Velocity of the SLR Station

Determining the velocities and direction of movement of the SLR stations relies on determining the coordinates for each observation epoch. By analyzing the linear regression of the components computed for the epoch of observations, their slope is determined, which is the velocity of the SLR station for a given component ($X$, $Y$, $Z$ or $N$, $E$, $U$). The straight line equation is as follows.

$$y = A + Bx \tag{5}$$

where:

$y$—$X$, $Y$, $Z$ or deviations from ITRF $N$, $E$, $U$ per reference epoch;

$x$—reference epoch;

$A$—systematic shift from zero;

$B$—slope (determined as a result of computing).

In order to precisely determine the line as a result of the linear regression analysis, the adopted observation period should not be shorter than three years (the longer the period, the more precise it is) because in the case of a shorter period the errors in determining the slope of the line significantly increase.

After obtaining the velocities of the coordinate components, we calculate the total velocity of the station motion *V3D*, which is the resultant of the three components, i.e., $V_X V_Y V_Z$ or $V_N V_E V_U$:

$$V3D = \sqrt{V_X{}^2 + V_Y{}^2 + V_Z{}^2} = \sqrt{V_N{}^2 + V_E{}^2 + V_U{}^2} \tag{6}$$

The total velocity *V3D* for the *XYZ* and *NEU* components should be equal; the permissible difference resulting from rounding is $\pm 0.2$ mm/year.

An important parameter for comparison and assessment of the quality of determination is the velocity of the station in the horizon plane *V2D* in which the movement of tectonic plates takes place.

$$V2D = \sqrt{V_N{}^2 + V_E{}^2} \tag{7}$$

To determine the direction of the station's movement, we use a simple azimuth formula (calculated from north to east). It indicates the direction of movement of a given tectonic plate at the location of the station.

$$A = arctg\left(\frac{V_E}{V_N}\right) \tag{8}$$

where, based on the velocity signs for directions *E* and *N*, the direction of movement of the tectonic plate is determined.

### 3. Network of the SLR Stations for the Period 2008–2013

First, the data of all the SLR stations in the period of study, i.e., from 1 January 2008 to 31 December 2012, which performed observations of the LAGEOS-1 and LAGEOS-2 satellites, were reviewed. The purpose of this review was to select those stations that will be subjected to a detailed analysis. Table A1 lists all the SLR stations (42) that performed observations in the selected period of five years, assuming that the number of normal points for selected SLR stations per a given monthly arc of the orbit is not less than 50 for the combined observations of both satellites. The established minimum limit of 50 normal points for the monthly arc is very important and results from the fact that with a smaller number of points, the error in determining the SLR station coordinates increases significantly, causing the determined positions to be burdened with a large error. The aforementioned limit is admittedly an arbitrary value, but it is supported by the experience of many SLR data analysis centers.

In summary, for the data set in Table A1, the normal points for the LAGEOS-1 and LAGEOS-2 satellites in the selected five-year period were taken from EDC (Eurolas Data Center) and subjected to initial processing, rejecting the results that did not meet the adopted minimum criterion 50 points normal for the monthly arc. Monthly arcs were introduced instead of weekly arcs because in this way a better accuracy in determining the station coordinates is obtained, due to the greater number of points and a more uniform distribution of stations in the monthly arc as compared to the weekly arc.

Under such conditions, the positions of the stations in the form of geocentric coordinates were computed for all stations, for which the average stability of the determined coordinates in the RMS form was computed for the entire observation period.

$$RMS_X = \sqrt{\frac{\sum_{i=1}^{n}\left(X_i - \overline{X}\right)^2}{n - 1}} \tag{9}$$

where *i* is the arc number, *X* is value of component (*X*, *Y*, *Z*) or (*N*, *E*, *U*) for each arc, $\overline{X}$ is the mean value of the components $X_i$, and *n* is a number of arcs. The total RMS for all the components (total station position stability) in reference to ITRF2008 was computed using Formula (10).

$$RMS3D = \sqrt{\frac{RMS_X^2 + RMS_Y^2 + RMS_Z^2}{3}} = \sqrt{\frac{RMS_N^2 + RMS_E^2 + RMS_U^2}{3}} \tag{10}$$

In order to assess the precision of the coordinate determination, the average 3D standard deviation was also computed for each arc.

## 4. Results

*4.1. Selection of SLR Stations*

The selection of SLR stations that conducted observations of the LAGEOS-1 and LAGEOS-2 satellites during the five-year period, i.e., from 1 January 2008 to 31 December 2012, for the analysis of determining the position and velocity of the stations, was carried out on the basis of the following criteria:

I. Number of normal points (N) of a given SLR station per one monthly arc of the orbit and covering the total observation of the LAGEOS-1 and LAGEOS-2 satellites for the entire observation period;

II. Observation period (Δ) for the SLR station data that must be contained in the range of not less than 3–5 years due to increasing errors in determining the velocity of the SLR station for shorter periods;

III. Stability of the determined coordinates (Equation (10)) in the absence of earthquakes is the best parameter to assess the accuracy of the observations of a given station.

The maximum number of monthly arcs for the assumed five-year period was 60, i.e., the station carried out measurements for 60 months. The selected 21 SLR stations are marked in bold in Table A1.

In summary, nine stations of the European network of EUROLAS stations qualified for the analysis: Concepcion (7405IR), Zimmerwald (7810), San Fernando (7824), Graz (7839), Herstmonceux (7840), Potsdam (7841), Grasse (7845), Matera (7941) and Wettzell (8834). From the West Pacific SLR Network (WPLTN), five SLR stations were included: Changchun (7237), Koganei (7308), San Juan (7406), Mount Stromlo (7825) and Simosato (7838). The remaining seven stations operate in the NASA network: McDonald (7080), Yarragadee (7090), Greenbelt (7105), Monument Peak (7110), Haleakala (7119), Tahiti (7124) and Hartebeesthoek (7501).

First, monthly arcs were rejected for stations whose sum of the normal points for the LAGEOS-1 and LAGEOS-2 satellites was less than 50. Then, after cleaning the remaining monthly data using the 3 sigma criterion, computations were carried out for each monthly arc using the orbital program GEODYN-II for the following parameters:

- Geocentric X, Y, Z and topocentric N, E, U [21] station coordinates for the reference epoch 2005.0 taking into account the velocity in the ITRF 2008—detrended data;
- Geocentric X, Y, Z and topocentric N, E, U station coordinates for reference epoch of each arc (first day of each month)—trended data;
- Standard deviations of the determined station coordinates;
- Stability (RMS) of designated X, Y, Z and N, E, U coordinates (9, 10);
- Total standard deviation (3D) of the coordinates (σ);
- Position uncertainty, i.e., error bars (3σ) resulting from the T-Student distribution;
- Number of normal points for each arc, separately for LEGEOS-1 and LAGEOS-2 satellites;
- Range bias for each arc, separately for LEGEOS-1 and LAGEOS-2 satellites;
- Orbital RMS for each arc, separately for LEGEOS-1 and LAGEOS-2 satellites;
- The number of RMS normal points for each arc for all the core stations;
- Orbital RMS for each arc for all the core stations.

*4.2. Station Position Changes*

The values of the components of the NEU station positions and their dispersion over the 5 years under study are presented in Table A2.

### 4.2.1. Earthquakes

The analysis of the obtained results showed, as expected, the greatest changes in the position and velocity of the two SLR stations near which the strong earthquakes occurred: Concepcion in Chile and Koganei in Japan. As a result of the earthquake of February 27 2010 (M8.8), the SLR station Concepcion moved 3219 mm for the E component,

673 mm for the N component and 45 mm for the U component (Figure 1). N, E, U are differences between determined station coordinates and ITRF2008 in Nord-South, East-West and vertical directions (blue squares). As a result of the earthquake of 11 March 2011 (Great Tohoku Earthquake—M9.1), the SLR station Koganei shifted the position for the E component by 418 mm, N by 56 mm and U by 13 mm (Figure 2). Both of these important SLR stations were closed in 2014 due to the poor quality of the results after the earthquake, preventing further analysis of the interesting changes in the positions of both stations after the earthquake. The effects of the Concepcion earthquake were detected at the SLR station in San Juan (Argentina), which is on the other side of the Andes Mountains. Shifts for this station occurred for all three components: in E by 32 mm, in N by 34 mm and in U an increase of 22 mm; a slight change in the station velocity was also found for the last two components (Figure 3). This SLR station was also closed in 2014. Several SLR stations changed position due to the 2011 earthquake in Japan. Position changes were found in the Simosato station (Japan) for the E component by 37 mm together with the change in velocity for this component (Figure 4). The Changchun station (China) also changed the position of the E component by 19 mm (Figure 5). An earthquake near the SLR station Monument Peak (California) on 3 April 2010 (M7.2) caused the station in the E component to shift by 21 mm and change the velocity of this component. All these results are fully consistent with the GNSS results [22,23]. Changes in position and velocity are of concern to these very important stations due to the quality of observations and their distribution on Earth, which resulted in a deterioration in the quality of determining satellite orbits, and thus in a deterioration in the quality of the determined parameters from the SLR observations.

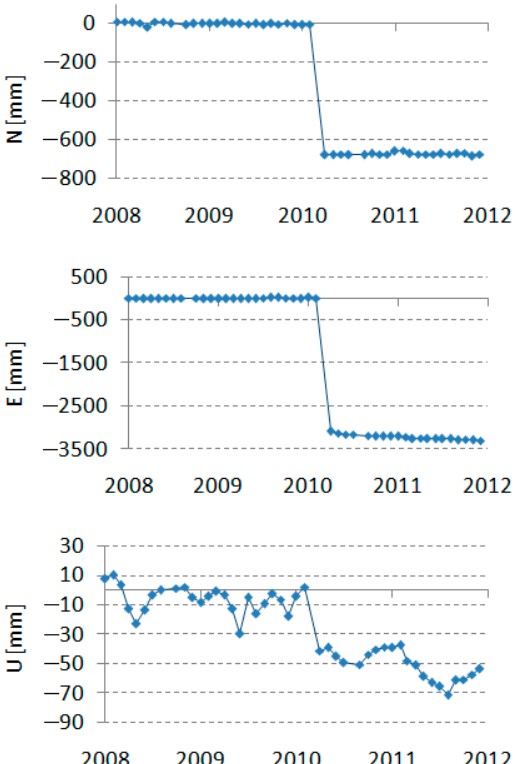

**Figure 1.** Changes in the NEU positions of the SLR station Concepcion (Chile) as a result of the earthquake on 27 February 2010.

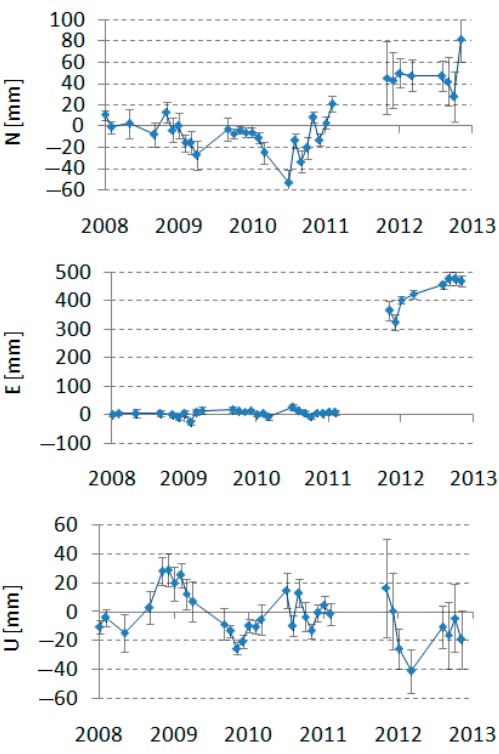

**Figure 2.** Changes in the NEU positions of the SLR station Koganei (Japan) as a result of the earthquake on 11 March 2011.

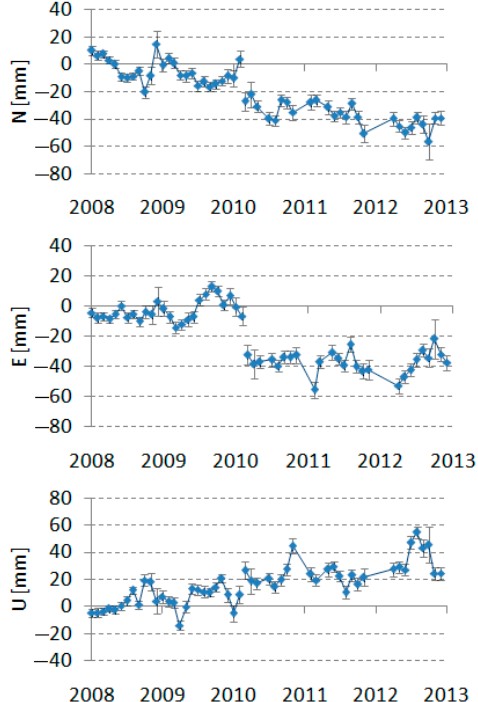

**Figure 3.** Changes in the NEU positions of the SLR station San Juan (Argentina) as a result of the earthquake in Chile on 27 February 2010.

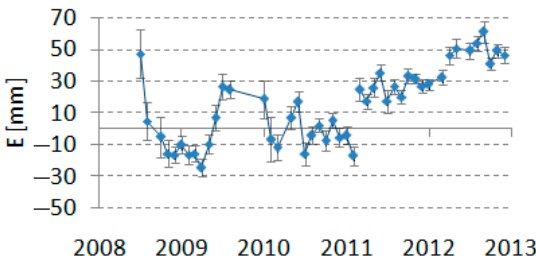

**Figure 4.** Change in component E of the SLR station Simosato (Japan) as a result of the earthquake on 11 March 2011.

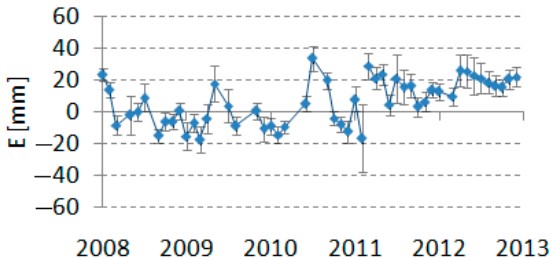

**Figure 5.** Change in component E of the SLR station Changchun (China) as a result of the earthquake in Japan on 11 March 2011.

### 4.2.2. Other Systematic Shifts

For two stations, McDonald (Texas) and Wettzell (Germany), jumps in the vertical component were found: for the SLR station McDonald in the period from June 2010 to March 2011 by +22 mm (Figure 6), and for the SLR station Wettzell from March 2009 to November 2010 by −30 mm (Figure 7). For the Potsdam SLR station (Germany) in October 2010, a jump in the vertical component of 11 mm was found (Figure 8). The results indicate a change in the bias of these stations. The SLR station Grasse (France) has a fixed vertical offset of 29 mm with respect to the ITRF2008, possibly due to an error in the vertical component of the ITRF. The SLR station Haleakala (Hawaii) shows a significant annual wave in the E and U components. For the E component, the annual wave amplitude is 2 cm, and for the U component, it is 2.5 cm (Figure 9); the semi-annual wave is also visible. These waves require an explanation, all the more because they also occur with similar amplitude in other periods. Removing all these biases and systematically checking the results would significantly improve the quality of the SLR results.

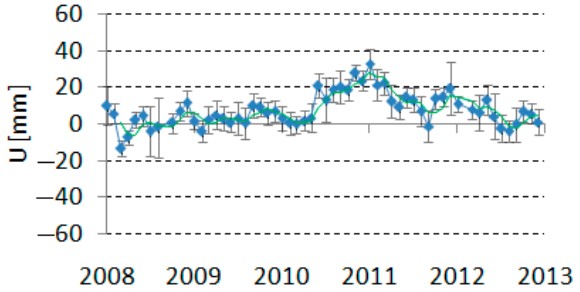

**Figure 6.** Change in the vertical component of the SLR station McDonald (Texas) due to station bias (the green line is the smoothed station coordinate).

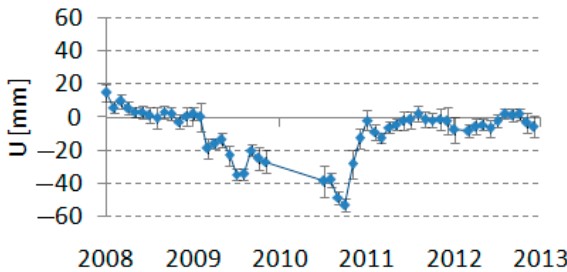

**Figure 7.** Change in the vertical component of the SLR station Wettzell (Germany) due to station bias.

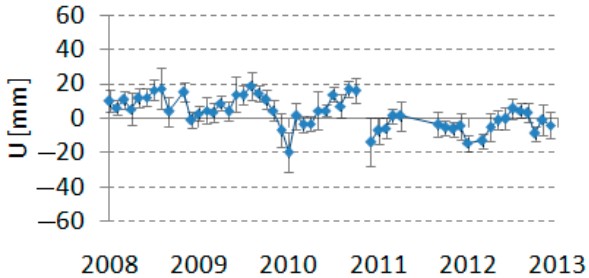

**Figure 8.** A jump in the vertical component of the SLR station Potsdam (Germany) in October 2010.

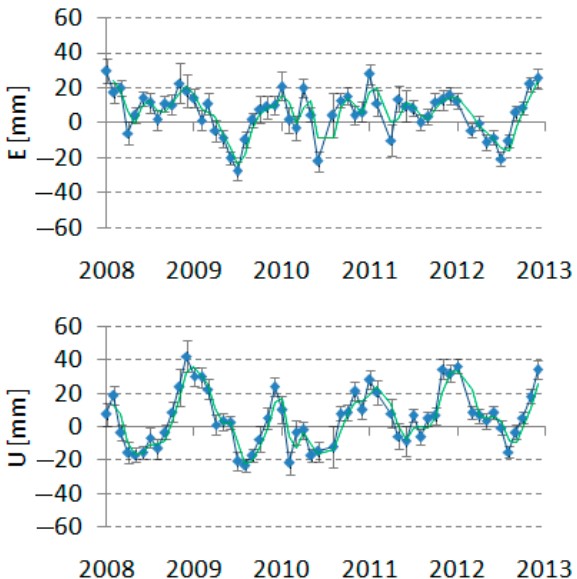

**Figure 9.** Annual wave at the E (**up**) and U (**down**) components of the SLR station Haleakala (Hawaii) (the green line is the smoothed station coordinate).

### 4.2.3. Random Errors

Several SLR stations show a significant increase in the random spread of station positions (RMSs) for individual components. The SLR station Monument Peak has a significantly larger spread for the U component ($\pm 15$ mm), the SLR station Tahiti for the N component ($\pm 14$ mm), the SLR station Hartbeesthoek (South Africa) for the E component ($\pm 13$ mm), and the SLR station San Fernando (Spain) for the E ($\pm 14$ mm) and U ($\pm 19$ mm) components. These significant increases in the spread of results compared to the standard spread for these stations on the level of $\pm 10$ mm should be explained.

The SLR stations Yarragadee (Western Australia), Greenbelt (Maryland), Zimmerwald (Switzerland), Graz (Austria), Herstmonceux (UK) and Matera (Italy) do not show any significant biases or random errors. An example of the results of the SLR station Zimmerwald is shown in Figure 10.

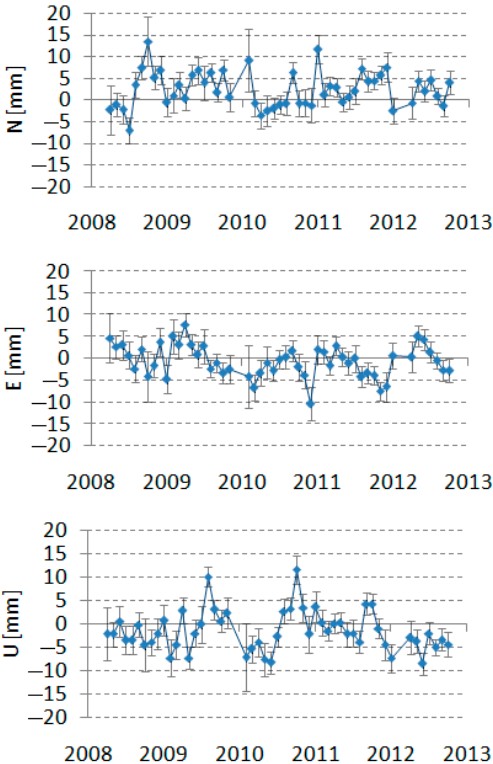

**Figure 10.** NEU components of the SLR station Zimmerwald (Switzerland)—detrended data.

*4.3. Station Velocities*

Referring to the principle of determining the SLR station velocity, in the next stage, in Tables 2 and A3, the data obtained as a result of the linear regression analysis for each of the 21 tested SLR stations are presented and interpreted.

**Table 2.** Linear regression analysis of topocentric components N and E for the reference epoch of observation for selected SLR stations that conducted observations in the period from 1 January 2008 to 31 December 2012.

| Station Number | a | Δa | $R^2$ | $S_y$ | F | $f$ | |
|---|---|---|---|---|---|---|---|
| 7080 | −6.6 | 0.8 | 0.5 | 9.2 | 60.1 | 56 | N |
| | −11.4 | 0.9 | 0.8 | 9.5 | 171.3 | 56 | E |
| 7090 | 56.7 | 0.4 | **1.0** | 4.0 | 24,712.2 | 57 | N |
| | 38.9 | 0.4 | **1.0** | 3.9 | 11,995.5 | 57 | E |
| 7105 | 4.0 | 0.5 | 0.6 | 5.5 | 64.9 | 50 | N |
| | −13.6 | 0.5 | **0.9** | 5.7 | 684.7 | 50 | E |
| 7110 | 15.1 | 0.9 | 0.8 | 9.5 | 284.9 | 54 | N |
| | −46.6 | 0.9 | **1.0** | 9.6 | 2717.9 | 54 | E |
| 7119 | 35.1 | 0.9 | **1.0** | 10.2 | 1457.7 | 55 | N |
| | −62.9 | 1.1 | **1.0** | 12.6 | 3025.7 | 55 | E |
| 7124 | 34.4 | 1.7 | **0.9** | 14.7 | 390.8 | 33 | N |
| | −67.1 | 1.3 | **1.0** | 11.1 | 2598.5 | 33 | E |
| 7237 all period | −14.8 | 1.0 | 0.8 | 10.6 | 215.3 | 49 | N |
| | 31.4 | 1.1 | **0.9** | 12.1 | 749.1 | 49 | E |

**Table 2.** *Cont.*

| Station Number | a | Δa | R² | Sy | F | ƒ | |
|---|---|---|---|---|---|---|---|
| 7237 to Feb. 2011 | −14.8 | 2.1 | 0.6 | 10.7 | 52.0 | 28 | N |
| | 25.4 | 2.5 | 0.8 | 13.0 | 103.4 | 28 | E |
| 7308 all period | 7.6 | 2.9 | 0.2 | 23.0 | 7.0 | 31 | N |
| | 104.6 | 13.4 | 0.7 | 107.1 | 61.3 | 31 | E |
| 7308 to Feb. 2011 | −9.1 | 3.4 | 0.2 | 15.7 | 7.2 | 23 | N |
| | 0.7 | 2.2 | 0.005 | 10.0 | 0.1 | 23 | E |
| 7405IR all period | 11.4 | 0.7 | 0.7 | 7.8 | 254.2 | 54 | N |
| | −199.6 | 14.9 | 0.8 | 162.2 | 180.1 | 54 | E |
| 7406 all period | 6.2 | 0.8 | 0.6 | 8.2 | 65.3 | 50 | N |
| | 0.5 | 1.2 | 0.003 | 12.4 | 0.2 | 50 | E |
| 7501 | 17.5 | 0.9 | **0.9** | 9.0 | 392.8 | 44 | N |
| | 15.7 | 1.2 | 0.8 | 12.4 | 165.3 | 44 | E |
| 7810 | 16.6 | 0.4 | **1.0** | 4.1 | 1453.7 | 49 | N |
| | 19.1 | 0.4 | **1.0** | 3.6 | 2589.0 | 49 | E |
| 7824 | 19.7 | 1.3 | **0.9** | 11.3 | 244.6 | 39 | N |
| | 16.0 | 1.6 | 0.7 | 14.5 | 97.9 | 39 | E |
| 7825 | 54.9 | 0.6 | **1.0** | 6.5 | 8650.0 | 56 | N |
| | 19.6 | 0.4 | **1.0** | 4.6 | 2210.7 | 56 | E |
| 7838 all period | 0.2 | 1.4 | 0.0004 | 12.5 | 0.018 | 44 | N |
| | 5.1 | 1.9 | 0.1 | 17.0 | 7.1 | 44 | E |
| 7838 to Feb. 2011 | −5.2 | 3.4 | 0.1 | 14.3 | 2.3 | 24 | N |
| | −9.6 | 4.1 | 0.2 | 16.9 | 5.6 | 24 | E |
| 7839 | 15.7 | 0.5 | **1.0** | 5.0 | 1196.1 | 57 | N |
| | 21.6 | 0.4 | **1.0** | 4.4 | 2960.7 | 57 | E |
| 7840 | 17.0 | 0.4 | **1.0** | 4.9 | 1464.6 | 57 | N |
| | 16.9 | 0.4 | **1.0** | 4.3 | 1930.0 | 57 | E |
| 7841 | 16.6 | 0.8 | **0.9** | 8.5 | 433.7 | 51 | N |
| | 17.4 | 0.6 | **0.9** | 6.5 | 825.1 | 51 | E |
| 7845 | 15.3 | 0.7 | **0.9** | 5.8 | 466.6 | 45 | N |
| | 20.5 | 0.7 | **1.0** | 5.7 | 867.3 | 45 | E |
| 7941 | 19.8 | 0.5 | **1.0** | 5.6 | 1456.3 | 53 | N |
| | 23.7 | 0.4 | **1.0** | 4.3 | 3551.9 | 53 | E |
| 8834 | 16.2 | 0.5 | **1.0** | 5.7 | 988.8 | 50 | N |
| | 20.2 | 0.5 | **1.0** | 6.1 | 1359.9 | 50 | E |

a—the slope of the regression line for N and E, the velocity of the station in a given component (mm/year); Δa—estimated mean error of the slope of the "a" coefficient (mm/year); R²—the square of the correlation coefficient, i.e., the coefficient of determination; Sy—mean error of the "y" value, i.e., the estimation of the standard deviation of the random component; F—Fisher's stat; ƒ—number of degrees of freedom of the residual variance.

The data in bold in the "R²" column list those SLR stations for which the quality of the regression equation is very good. The pre-earthquake results in Japan for three stations, Koganei, Simosato and Changchun, are separately shown in the table, which better reflects the situation of these stations.

When analyzing the obtained data presented in Table 2, it was found that the variability of Y was explained in 100% via the regression model for the topocentric coordinates N and E for the following SLR stations: Yarragadee (7090), Haleakala (7119), Zimmerwald (7810), Mount Stromlo (7825), Graz (7839), Herstmonceux (7840), Matera (7941) and Wettzell (8834). The stations of Tahiti (7124), Potsdam (7841) and Grasse (7845) can also be included in this group, with the difference that for one or two coordinates, the variability of Y was explained in 90% via the regression model. The worst in this ranking are the stations where an earthquake was recorded in the period under review, i.e., Changchun (7237), Koganei (7308), Concepción (7405IR), San Juan (7406) and Simosato (7838). For example, for the Simosato SLR station and the entire five-year period under consideration, the regression model for the topocentric coordinate N explained the variability of Y only in 0.042%, i.e., in this case, the identity factor ($\varphi2$) is as much as 99.96%, which also translates into a small result of the Fisher statistical criterion: 0.018. Moreover, very high values for the smallest average error of the estimation of the slope of the coefficient "a" (component velocities) were recorded for the topocentric coordinate E of the following SLR stations: Koganei—the entire observation period ($\Delta a$ = 13.4 mm/year) and Concepción—the entire observation period ($\Delta a$ = 14.9 mm/year). On the other hand, the smallest mean estimated slope error for N and E of the "a" coefficient was recorded for the following stations: Yarragadee ($\Delta a$ = 0.4 mm/year), Zimmerwald ($\Delta a$ = 0.4 mm/year) and Herstmonceux ($\Delta a$ = 0.4 mm/year).

For the analysis of changes in the position of the station after the earthquake, the strongest quakes in Concepcion and Koganei were taken into account. The results of the changes in the position of the Concepcion station before and after the earthquake are shown in Figure 11. Similar results were obtained for the Koganei station (Figure 12), but the velocities for this station were almost zero before the earthquake (Table A3). It is noteworthy that the position of both stations after the earthquake is quadratic rather than linear, which causes a constant change in the direction and velocity of the station. This is the result of the recovery of the area in the area of the quake and a gradual return to the situation before the earthquake, which in the case of a strong earthquake may take even several years (constant change in direction and velocity). Finally, both Concepcion and Koganei stations were closed in 2014. Similar changes for the NASA SLR station in Arequipa (Peru), which occurred as a result of the 2001 earthquake, are presented in the paper [22].

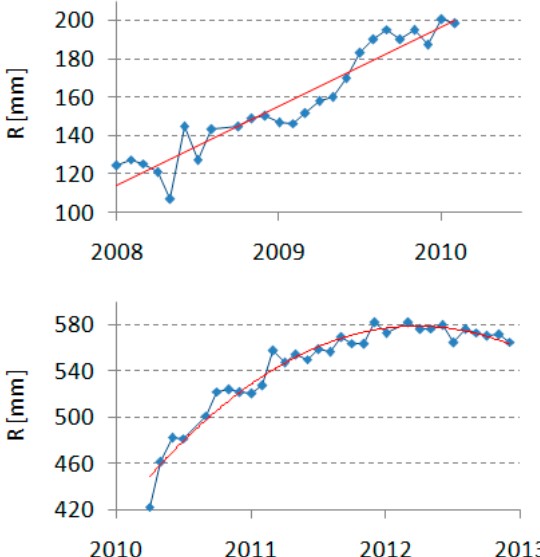

**Figure 11.** The resultant NEU components before (**up**) and after the earthquake (**down**) of the SLR station Concepcion (Chile) on 27 February 2010—trended data (blue squares are resultant of the station position in comparison to ITRF2008 and red line is the linear (**up**) and square (**down**) alignment of the positions).

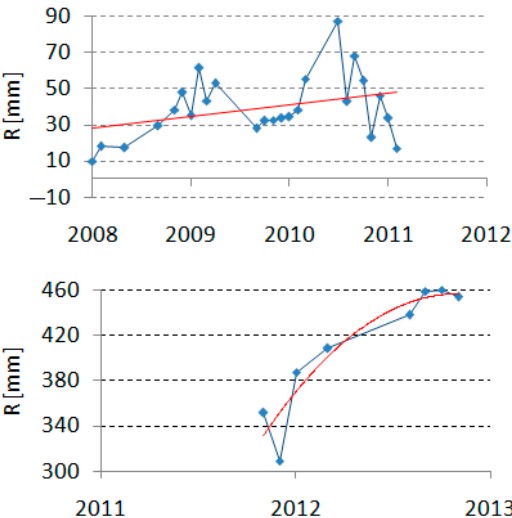

**Figure 12.** The resultant NEU components before (**up**) and after the earthquake (**down**) of the SLR station Koganei (Japan) on 11 March 2011—trended data (blue squares are resultant of the station position in comparison to ITRF2008 and red line is the linear (**up**) and square (**down**) alignment of the positions).

## 5. Discussion and Conclusions

The aim of this work, as indicated in the introduction, was to analyze the SLR observations for determining changes in the position and velocity of the SLR stations in the period under study, especially after very strong earthquakes, and to assess the quality of the individual stations, both from the point of view of systematic and random errors. For this purpose, a particularly interesting period of five years, 2008.0–2013.0, was selected, during which two strong earthquakes occurred that impacted the SLR stations important for the orbit determination process. The study found that only 6 out of 42 SLR stations operating in this period had no significant changes in position during the 5 years under study and their observations were of high quality. As many as half of all the stations (21) were not qualified for the analysis, mainly due to too large gaps in the observations, less number of such stations and insufficient accuracy and precision of the observations.

Determining the position and velocity of the SLR stations, as well as observing their changes over the years, is mainly used to illustrate changes in the coordinates of individual points on the Earth's surface, which translates into tracking the causes of these changes, i.e., the movement of tectonic plates, earthquakes and systematic errors. Furthermore, in the SLR technique, it is sought that in the future, according to the Global Geodetic Observing System (GGOS) assumptions, these parameters will be determined with an accuracy of 1 mm for position and 0.1 mm/year for the velocity of the tested SLR stations.

For most of the observed period, the changes in individual topocentric NEU components of the 21 analyzed SLR stations were relatively small (a few millimeters). However, the largest and smallest recorded variability in relation to the ITRF2008 for the 2005.0 epoch without earthquake effects are presented below.

- It was found that the largest jump in the component N for the examined period of five years was recorded for the Koganei station (7308), amounting to 39.3 mm (July 2010), computed for the period before the earthquake. On the other hand, the smallest noticeable shift was 9.5 mm for the Yarragadee station (7090) (May 2009). It can therefore be concluded that the noticeable variability in the component N of the analyzed SLR stations oscillated between 9.5 mm and 39.3 mm;
- The largest jump in the component E for the analyzed period was recorded for the San Fernando station (7824), which was 55.0 mm (January 2008). The smallest observable change was recorded for the Zimmerwald station (7810) and it amounted to 12.3 mm (December 2010). It should therefore be concluded that the noticeable variation in

component E of the positions of the tested SLR stations ranges from 12.3 mm to 55.0 mm;

- The largest visible jump for the vertical component U was 31.7 mm (May 2008) at the Hartebeesthoek station (7501), while the smallest visible jump was 9.4 mm (June 2009) at the Herstmonceux station (7840). Thus, the variability in the vertical component U for the analyzed SLR stations in the five-year period oscillated between 9.4 mm and 31.7 mm.

Generally, for all three components, the changes in position are similar, at the level of several mm.

### 5.1. Changes in Position Caused by Earthquakes

In the case of earthquakes, the shifts are much larger in some stations: Koganei (7308), Simosato (7838) and Changchun (7237) (earthquake off the coast of Japan on 11 March 2011), and Concepcion and San Juan (7406) (earthquake in Chile on 27 February 2010). Stabilization of the position of Concepcion (change in E by 3 m) and Koganei (change in E by 40 cm) stations would take place only after a dozen or so years; for the remaining stations, the changes are at the level of several cm and do not have a significant impact on the results of these stations.

The change in the position of the stations as a result of the movement of tectonic plates takes place only in the horizontal direction, while at the time of an earthquake within the area under study, a significant difference may also occur in the vertical component. This statement is confirmed by the data obtained for the component U (vertical) relating to the SLR stations where the earthquake occurred: in the Concepcion station (the change in the vertical component U was $-50.4$ mm at the time of the quake, a very significant change) and in the Koganei station (the change in the component U at the time of the quake was $-19.2$ mm). For the other listed SLR stations, the changes in the vertical component after the earthquake are much smaller and do not exceed 20 mm: San Juan (18.1 mm), Simosato (3.3 mm), and Changchun (14.7 mm). The change in the vertical component for the Monument Peak station (7110) as a result of the earthquake amounted to 4.4 mm and is not significant. Thus, only with very strong earthquakes there is a significant change in the vertical position of the station. In addition to the earthquake phenomenon described above, the actual change in the vertical position of the SLR stations can also be influenced by Earth tidal effects, post-glacial uplifts, as well as regional and local movements.

### 5.2. Changes in Position Caused by Systematic Errors

Vertical changes in position also reflect the systematic errors of the station, which are most visible in the direction of observation, i.e., vertical. During the period under study, such errors were detected for several very good stations: errors lasted for some time in McDonald (7080) and Wettzell (8834), and there was a constant jump in component U by 30 mm in the Potsdam station (7841). One SLR station (Haleakala (7119)) had a strong annual wave and a visible semi-annual wave in the E and U components. Systematic shift in results due to fixed errors in ITRF2008 have been registered for several SLR stations (Changchun (7237) by 22 mm in component U, Simosato (7838) by 36 mm in component N and 50 mm in component U and Grasse by 29 mm in component U). The quick elimination of these errors should be an important task to improve the quality of SLR station results.

### 5.3. Station Velocities

The analyzed velocities of the individual 21 SLR stations were characterized by very good stability in most cases. European SLR stations stood out especially in this respect: Zimmerwald (7810), Graz (7839), Herstmonceux (7840), Potsdam (7841), Grasse (7845), Matera (7941) and Wettzell (8834), and Australian SLR stations as well: Yarragadee (7090) and Mount Stromlo (7825). The exceptions were the stations where the earthquake phenomenon was recorded and which, therefore, were characterized by rather poor stability of individual velocities. This relationship results from the fact that at the time of an earthquake, the posi-

tion and velocity of the SLR stations change, whose stabilization due to the "rebounding" of the earthquake will take place only after a dozen or so years. It should be noted that after the strong earthquake that was recorded in Concepcion and Koganei, the position of the station, as presented in the paper, underwent not linear but quadratic changes. It is difficult to determine how long such changes would last due to the closure of both stations in 2014. The only station that continues observations after the strong earthquake (23 June 2001) is Arequipa in Peru, whose results (station velocity and direction) changed for several years [22].

### 5.4. Accuracy

Observation accuracy determined as the average stability of the determined 3DRMS coordinates (Sc) ranges from 4.0 mm to 6.5 mm for the best stations (8 stations) and from 8.2 mm to 17.5 mm for stations with worse results (13 stations). The precision of observations in the 5 years under study, measured via the average standard deviation of station coordinates ($\sigma$) for the 21 analyzed SLR stations, was from 0.9 mm to 1.7 mm for the best SLR stations and from 1.9 mm to 3.4 mm for SLR stations that had worse results. In addition, the stability of the individual components of the NEU coordinates for the five-year period ranges from 4.1 mm (Zimmerwald) to 15.8 mm (Koganei) for component N, from 3.6 mm (Zimmerwald) to 16.7 mm (Simosato) for component E and from 4.2 mm (Graz, Herstmonceux) to 21.1 mm (Simosato) for component U. Summing up, the stations Zimmerwald (7810), Herstmonceux (7840) and Graz (7839) performed best in the period under study.

Further work related to improving the quality of SLR stations results and achieving the GGOS level (1 mm in position and 0.1 mm/year in velocity) should focus on improving the tropospheric correction, either via two-color observations [24] or by introducing corrections for horizontal gradients [25–27]. Differences in the quality and quantity of observations, which, with the exception of six stations, show notable jumps in the results, require significant improvement. Continuous monitoring of the observation results in the form of determined stations coordinates from the observations of the LAGEOS-1 and LAGEOS-2 satellites, and in the future if extended to include other satellites, would make it possible to avoid the systematic errors presented in this work and thus would increase the number of stations whose results form the basis for determining the orbits via the SLR technique.

### 5.5. Conclusions

The authors believe the most important contribution of this work is the demonstration of a quadratic change in the position of the station after strong earthquakes. This is shown in the lower part of Figures 11 and 12. This allows for determining the position and velocity of the station at arbitrary moment after the earthquake, and thus it is easier to remove the effects of strong earthquakes compared to the one used in ITRF2014 and ITRF2020 post-seismic deformation model (PSD).

Another important information obtained in this work is the detected jumps of 2–3 cm for three SLR stations; Chanchun, Simosato (as an effect of the tsunami in Japan) and San Juan (as an effect of the Concepcion earthquake). These jumps can be easily eliminated by the ITRF in the form of additional station coordinates before and after the earthquake (this is implemented in the ITRF). The systematic errors found in this work for the three stations can be or were eliminated by introducing appropriate range biases. Finally, the annual waves for the Haleakala station can be removed in ITRF2020 by entering the annual and semi-annual frequencies.

The most important issue is that the information about the observed changes in the location of the station should be monitored, for example, in the form presented in this paper, and transmitted to the station as soon as possible. Such actions would improve the quality of the SLR stations, and thus bring the quality of SLR stations closer to the primary goal of the Global Geodetic Observing System (GGOS) of achieving a positioning accuracy of 1 mm and a velocity of 0.1 mm/year.

**Author Contributions:** Conceptualization, S.S. and A.S.; methodology, S.S.; software, P.M. and S.S.; validation, D.S., A.S. and S.S.; formal analysis, D.S.; investigation, S.S. and A.S.; resources, P.M. and A.S.; data curation, S.S.; writing—original draft preparation, S.S., A.S. and D.S.; writing—review and editing, D.S, A.S. and S.S.; visualization, D.S. and P.M.; supervision, S.S.; project administration, S.S.; funding acquisition, S.S. All authors have read and agreed to the published version of the manuscript.

**Funding:** This research received no external funding.

**Data Availability Statement:** Input SLR data are available at the open access EUROLAS Data Center (EDC).

**Acknowledgments:** The authors wish to thank the Goddard Space Flight Centre National Aeronautics and Space Administration for the permission to use the GEODYN-II orbital software and thank the International Laser Ranging Service stations for their continuous effort in providing high-quality satellite laser ranging data.

**Conflicts of Interest:** The authors declare no conflict of interest.

## Appendix A

**Table A1.** List of SLR stations that performed observations of the LAGEOS-1 and LAGEOS-2 satellites in the period from 1 January 2008 to 31 December 2012.

| Station Number | Station Name | Period of Observation Monthyear | Time of Observations in Years ($\Delta$) | Number of Monthly Arcs (N) | Mean Stability of Station Coordinates 3D [mm] (Sc) | Mean Standard Deviation of the Station Coordinates Determination [mm] ($\sigma$) |
|---|---|---|---|---|---|---|
| 1824 | Kyiv Ukraine | 0810–1210 | 4.1 | 17 | 28.2 | 4.5 |
| 1868 | Komsomolsk-na-Amure Russia | 0812–1212 | 4.1 | 7 | 35.1 | 11.2 |
| 1873 | Simeiz Crimea | 0801–1212 | 5.0 | 24 | 49.0 | 4.1 |
| 1879 | Altay Russia | 0810–1212 | 4.2 | 31 | 17.8 | 4.0 |
| 1884 | Riga Latvia | 0805–1111 | 3.4 | 25 | 25.7 | 4.7 |
| 1886 | Arkhyz Russia | 1103–1211 | 1.7 | 12 | 20.6 | 4.2 |
| 1887 | Baikonur Kazakhstan | 1204–1211 | 0.7 | 8 | 11.5 | 2.6 |
| 1888 | Svetloe St.Petersburg Russia | 1205 | - | 1 | - | 3.1 |
| 1889 | Zelenchukskya Russia | 1204–1210 | 0.6 | 6 | 13.7 | 5.0 |
| 1890 | Badary Russia | 1203–1210 | 0.7 | 7 | 12.6 | 4.1 |
| 1893 | Katzively Crimea | 0802–1212 | 4.9 | 51 | 18.8 | 4.1 |
| **7080** | McDonald Texsas-USA | 0801–1212 | 5.0 | 58 | 9.3 | 2.4 |
| **7090** | Yarragadee West Australia | 0801–1212 | 5.0 | 59 | 5.0 | 0.9 |
| **7105** | Greenbelt Maryland-USA | 0801–1212 | 5.0 | 52 | 5.9 | 1.7 |

**Table A1.** *Cont.*

| Station Number | Station Name | Period of Observation Monthyear | Time of Observations in Years (Δ) | Number of Monthly Arcs (N) | Mean Stability of Station Coordinates 3D [mm] (Sc) | Mean Standard Deviation of the Station Coordinates Determination [mm] (σ) |
|---|---|---|---|---|---|---|
| 7110 | Monument Peak California-USA | 0801–1212 | 5.0 | 56 | 12.8 | 1.8 |
| 7119 | Haleakala Hawaii-USA | 0801–1212 | 5.0 | 57 | 13.5 | 1.9 |
| 7124 | Tahiti French Polynesia | 0806–1211 | 4.5 | 35 | 12.0 | 2.8 |
| 7237 | Changchun China | 0801–1212 | 5.0 | 51 | 13.9 | 2.4 |
| 7249 | Beijing China | 0801–1210 | 4.8 | 26 | 20.9 | 3.6 |
| 7308 | Koganei Tokyo-Japan | 0801–1211 | 4.9 | 33 | 13.9 | 2.8 |
| 7328 | Koganei Tokyo-Japan | 1012–1102 | 0.2 | 3 | 4.5 | 3.6 |
| 7358 | Tanegashima Kiusiu-Japan | 0804–1104 | 3.1 | 18 | 17.4 | 4.1 |
| 7403 | Arequipa Peru | 0806–1211 | 4.5 | 23 | 13.2 | 2.8 |
| 7405B | Concepcion Chile | 0909–0911 | 0.2 | 3 | 3.3 | 2.6 |
| 7405IR | Concepcion Chile | 0801–1212 | 5.0 | 56 | 21.8 | 1.7 |
| 7406 | San Juan Argentina | 0801–1212 | 5.0 | 52 | 17.4 | 1.6 |
| 7501 | Hartebeesthoek Pretoria-South Africa | 0802–1212 | 4.9 | 46 | 13.1 | 2.1 |
| 7810 | Zimmerwald Switzerland | 0804–1210 | 4.6 | 51 | 4.0 | 1.0 |
| 7811 | Borowiec Poland | 0806–0909 | 1.2 | 7 | 12.2 | 7.5 |
| 7820 | Kunming Junan-China | 0804–0902 | 0.9 | 3 | 33.9 | 5.0 |
| 7821 | Shanghai China | 0805–1210 | 4.6 | 30 | 13.6 | 3.6 |
| 7822 | Tahiti French Polynesia | 1106–1109 | 0.3 | 4 | 8.1 | 2.7 |
| 7824 | San Fernando Spain | 0801–1210 | 4.8 | 41 | 15.3 | 3.4 |
| 7825 | Mount Stromlo Canberra-Australia | 0801–1212 | 5.0 | 58 | 6.3 | 1.6 |
| 7832 | Riyadh Saudi Arabia | 0801–1110 | 3.8 | 33 | 8.9 | 2.5 |
| 7838 | Simosato Honsiu-Japan | 0807–1212 | 4.5 | 46 | 19.2 | 2.2 |
| 7839 | Graz Austria | 0801–1212 | 5.0 | 59 | 4.5 | 1.6 |
| 7840 | Herstmonceux UK | 0801–1212 | 5.0 | 59 | 4.4 | 1.4 |

**Table A1.** *Cont.*

| Station Number | Station Name | Period of Observation Monthyear | Time of Observations in Years ($\Delta$) | Number of Monthly Arcs (N) | Mean Stability of Station Coordinates 3D [mm] (Sc) | Mean Standard Deviation of the Station Coordinates Determination [mm] ($\sigma$) |
|---|---|---|---|---|---|---|
| **7841** | Potsdam Germany | 0801–1212 | 5.0 | 53 | 8.4 | 2.2 |
| **7845** | Grasse France | 0811–1212 | 4.1 | 47 | 6.3 | 1.7 |
| **7941** | Matera Italy | 0801–1212 | 5.0 | 55 | 5.5 | 1.2 |
| **8834** | Wettzell Germany | 0801–1212 | 5.0 | 52 | 9.8 | 1.6 |

Notice: Observations conducted in Concepcion in blue (7405B) and in infrared (7405IR) were treated as separate stations. In addition to the laser stations listed in Table A1, LAGEOS satellite observations were also carried out in this period by several other stations: Lviv (1831), Maidanak (1864), Burnie (7370), Ajaccio (7848), Zimmerwald (7810B, 7810IR), however, the number of normal points per month with the exception of Zimmerwald (7810 Blue and Infrared—only one month of observations), did not exceed 50 and therefore are not included in this table. In bold—selected 21 SLR stations.

**Table A2.** List of designated NEU positions in relation to ITRF2008 for selected 21 SLR stations—detrended data.

| SLR Station | N [mm] | RMS for N [mm] | E [mm] | RMS for E [mm] | U [mm] | RMS for U [mm] | 3DRMS [mm] | 3D$\sigma$ [mm] |
|---|---|---|---|---|---|---|---|---|
| McDonald (7080) | 2.3 | 9.2 | −2.8 | 9.6 | 7.5 | 9.0 | 9.3 | 2.4 |
| Yarragadee (7090) | −1.1 | 4.4 | 2.1 | 3.9 | −1.9 | 6.4 | 5.0 | 0.9 |
| Greenbelt (7105) | 3.2 | 5.4 | −1.5 | 5.9 | 5.4 | 6.4 | 5.9 | 1.7 |
| Monument Peak before EQ (7110) | −6.1 | 10.0 | **−1.0** | 11.2 | −6.3 | 15.1 | 12.3 | 2.2 |
| Monument Peak after EQ (7110) | −12.1 | 9.9 | **−22.4** | 9.8 | −11.7 | 10.0 | 9.9 | 1.5 |
| Haleakala (7119) | −0.7 | 10.3 | 5.5 | 12.6 | 5.0 | 16.7 | 13.5 | 1.9 |
| Tahiti (7124) | −2.4 | 14.5 | 1.6 | 11.1 | −0.3 | 9.9 | 12.0 | 2.8 |
| Changchun before EQ (7237) | −3.2 | 10.8 | **−1.4** | 12.8 | 22.3 | 13.8 | 12.5 | 2.3 |
| Changchun after EQ (7237) | −9.5 | 11.1 | **17.3** | 7.1 | 37.4 | 13.8 | 11.5 | 2.5 |
| Koganei before EQ (7308) | **−8.5** | 15.8 | **4.6** | 10.1 | 0.2 | 15.0 | 13.9 | 2.8 |
| Koganei after EQ (7308) | **47.8** | 15.0 | **423.4** | 56.5 | −12.5 | 17.6 | 35.2 | 7.1 |
| Concepcion before EQ (7405IR) | **−0.4** | 6.6 | **−2.4** | 10.1 | **−6.2** | 9.2 | 8.8 | 1.4 |
| Concepcion after EQ (7405IR) | **−673.2** | 6.8 | **−3221.1** | 56.7 | **−50.7** | 10.3 | 33.5 | 30.3 |
| San Juan before EQ (7406) | **−4.8** | 9.0 | **−2.9** | 7.0 | **5.0** | 8.6 | 8.2 | 1.4 |
| San Juan after EQ (7406) | **−37.2** | 8.6 | **−37.1** | 7.4 | **27.0** | 11.1 | 9.2 | 1.8 |

**Table A2.** *Cont.*

| SLR Station | N [mm] | RMS for N [mm] | E [mm] | RMS for E [mm] | U [mm] | RMS for U [mm] | 3DRMS [mm] | 3Dσ [mm] |
|---|---|---|---|---|---|---|---|---|
| Hartebeesthoek (7501) | −3.2 | 9.0 | −3.2 | 12.9 | 3.0 | 11.4 | 11.2 | 2.1 |
| Zimmerwald (7810) | 2.4 | 4.1 | −0.6 | 3.6 | −1.5 | 4.3 | 4.0 | 1.0 |
| San Fernando (7824) | 8.6 | 11.8 | −2.1 | 14.4 | 6.0 | 18.8 | 15.3 | 3.4 |
| Mount Stromlo (7825) | −3.0 | 6.9 | 2.7 | 5.2 | 2.2 | 6.7 | 6.3 | 1.6 |
| Simosato before EQ (7838) | −36.4 | 14.1 | **−1.0** | 16.7 | −49.6 | 21.1 | 17.5 | 2.5 |
| Simosato after EQ (7838) | −24.4 | 8.1 | **35.8** | 12.9 | −40.3 | 15.0 | 12.3 | 1.7 |
| Graz (7839) | 4.5 | 5.0 | 1.2 | 4.3 | 2.6 | 4.2 | 4.5 | 1.6 |
| Herstmonceux (7840) | 5.0 | 4.9 | −0.3 | 4.2 | −2.2 | 4.2 | 4.4 | 1.4 |
| Potsdam (7841) | 2.3 | 8.7 | 0.3 | 7.0 | 2.8 | 9.2 | 8.4 | 2.2 |
| Grasse (7845) | 1.8 | 5.9 | 3.0 | 5.6 | 28.8 | 7.3 | 6.3 | 1.7 |
| Matera (7941) | 4.6 | 5.7 | −2.0 | 4.2 | 0.7 | 6.5 | 5.5 | 1.2 |
| Wettzell (8834) | −3.5 | 5.6 | −2.7 | 6.0 | −8.8 | 14.9 | 9.8 | 1.6 |

Notice: The NEU components for the stations after the strong earthquakes in Concepcion and Koganei are approximate values as averages over the period after the quake in fact the positions of the NEU stations are quadratic variables, not linear as in the case of other stations. Significant jumps in station components are marked in bold.

**Table A3.** List of designated velocities for selected SLR stations per epoch of observation for the period from January 2008 to December 2012.

| Nr | SLR Station | Component Velocities [mm/year] | | | | | | Total Velocities [mm/year] | | | |
|---|---|---|---|---|---|---|---|---|---|---|---|
| | | $V_X$ | $V_Y$ | $V_Z$ | $V_N$ | $V_E$ | $V_U$ | V3D (XYZ) | V3D (NEU) | V2D (NE) | Azimuth [deg.] |
| 1 | McDonald (7080) | −12.2 | −1.8 | −4.8 | −6.6 | −11.4 | 1.6 | 13.2 | 13.3 | 13.2 | 239.9 |
| 2 | Yarragadee (7090) | −46.5 | 7.5 | 49.8 | 56.7 | 38.9 | −0.9 | 68.6 | 68.8 | 68.8 | 34.5 |
| 3 | Greenbelt (7105) | −13.7 | −1.1 | 3.4 | 4.0 | −13.6 | 0.5 | 14.2 | 14.2 | 14.2 | 286.4 |
| 4 | Monument Peak (7110) | −37.3 | 29.8 | 11.3 | 15.1 | −46.6 | −2.3 | 49.1 | 49.0 | 49.0 | 288.0 |
| 5 | Haleakala (7119) | −14.9 | 62.2 | 33.0 | 35.1 | −62.9 | 1.0 | 72.0 | 72.0 | 72.0 | 299.2 |
| 6 | Tahiti (7124) | −42.5 | 52.8 | 32.8 | 34.4 | −67.1 | −0.4 | 75.3 | 75.4 | 75.4 | 297.1 |
| 7 | Changchun (7237) | −32.4 | −8.7 | −9.1 | −14.8 | 31.4 | 2.1 | 34.8 | 34.8 | 34.7 | 115.2 |
| 8 | Koganei (7308) to Feb. 2011 | −4.0 | 2.5 | −7.8 | −9.1 | 0.7 | −0.8 | 9.1 | 9.1 | 9.2 | 175.6 |

**Table A3.** *Cont.*

| Nr | SLR Station | Component Velocities [mm/year] | | | | | | Total Velocities [mm/year] | | | |
|---|---|---|---|---|---|---|---|---|---|---|---|
| | | $V_X$ | $V_Y$ | $V_Z$ | $V_N$ | $V_E$ | $V_U$ | V3D (XYZ) | V3D (NEU) | V2D (NE) | Azimuth [deg.] |
| 9 | Concepcion (7405IR) to Feb. 2010 | 42.1 | 10.1 | 15.4 | 16.4 | 38.5 | −4.7 | 46.0 | 42.1 | 41.8 | 66.9 |
| | Concepcion (7405IR) from Apr. 2010 | −37.1 | −14.7 | 18.4 | 16.7 | −39.8 | −8.5 | 43.9 | 44.0 | 43.2 | 292.8 |
| 10 | San Juan (7406) to Feb. 2010 | 17.9 | −4.6 | 6.1 | 10.9 | 14.9 | 6.0 | 19.5 | 19.4 | 18.5 | 53.8 |
| | San Juan (7406) from Mar. 2010 | 13.5 | −6.7 | 5.6 | 10.7 | 10.2 | 6.6 | 16.1 | 16.2 | 14.8 | 43.6 |
| 11 | Hartebeesthoek (7501) | 0.5 | 17.8 | 15.1 | 17.5 | 15.5 | 1.3 | 23.3 | 23.4 | 23.4 | 41.5 |
| 12 | Zimmerwald (7810) | −13.7 | 17.4 | 12.2 | 16.6 | 19.1 | 1.2 | 25.3 | 25.3 | 25.3 | 49.0 |
| 13 | San Fernando (7824) | −9.1 | 17.1 | 16.4 | 19.7 | 16.0 | 1.0 | 25.4 | 25.4 | 25.4 | 39.1 |
| 14 | Mount Stromlo (7825) | −35.8 | −1.4 | 45.8 | 54.9 | 19.6 | −2.1 | 58.1 | 58.3 | 58.3 | 19.6 |
| 15 | Simosato (7838) to Feb. 2011 | 7.0 | 6.5 | −6.5 | −5.2 | −9.6 | −4.1 | 11.6 | 11.7 | 10.9 | 118.4 |
| 16 | Graz (7839) | −16.6 | 17.8 | 11.0 | 15.7 | 21.6 | 0.4 | 26.7 | 26.7 | 26.7 | 54.0 |
| 17 | Herstmonceux (7840) | −13.7 | 16.8 | 10.2 | 17.0 | 16.9 | −0.6 | 24.0 | 24.0 | 24.0 | 44.8 |
| 18 | Potsdam (7841) | −18.8 | 13.4 | 7.3 | 16.6 | 17.4 | −3.5 | 24.2 | 24.3 | 24.0 | 46.3 |
| 19 | Grasse (7845) | −13.9 | 19.0 | 10.1 | 15.3 | 20.5 | −1.4 | 25.6 | 25.6 | 25.6 | 53.3 |
| 20 | Matera (7941) | −19.1 | 19.1 | 15.1 | 19.8 | 23.7 | 0.1 | 30.9 | 30.9 | 30.9 | 50.1 |
| 21 | Wettzell (8834) | −16.0 | 17.1 | 11.1 | 16.2 | 20.2 | 0.6 | 25.9 | 25.9 | 25.9 | 51.3 |

Notice: for the SLR stations Koganei and Simosato, the post-quake period is too short to determine the correct station velocity, for the SLR station San Juan, pre- and post-quake results are shown due to significant jumps in all three components after the Chile earthquake. This very good station also stopped observations in 2014. For the remaining stations affected by the earthquake, the velocity changes are too small, and therefore the results for the entire five-year period are given.

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
