# Peer review of "Analysis of the Results Determining the Positions and Velocities of Satellite Laser Ranging Stations during Earthquakes in 2010–2011"

_remotesensing, doi:10.3390/rs15143659_

Round 1

Reviewer 1 Report

The paper sets out to discuss the geodetic use of laser ranging observations from the global network from a five-year period during which two major earthquakes took place. The paper's stated aim is to monitor the geocentric positions and velocities of two of the stations most affected by the earthquakes, as well as an assessment of the quality of the coordinates retrieved for all the stations used in the analysis.

An overall impression of the paper is that not enough detail is given about the force model and station displacement models used by the GEODYN-II analysis package. The paper also feels very much 'of of date' with work ignored on improved SLR analyses in recent years; for instance no reference is made to improved models for centre-of-mass corrections for the geodetic satellites (Rodriguez et al, 2019) and the ILRS Analysis Standing Committee work on mitigation of known systematic errors in the stations (Luceri, et al, 2019), now adopted by all the Analysis Centres and used for the ILRS input to the new version of the reference frame (ITRF2020, Altamimi et al, 2023). There is also a newer reference for the ILRS (Pearlman et al, 2019).

To ignore these improvements in data handling means that many of the conclusions about station stability are not relevant and very misleading to the reader and users of the SLR technique.

The motions of the two targeted stations as affected by the earthquakes are well-described and useful results from the technique. But no reference at all is made to the work presented in the ITRF 2014 and ITRF 2020 realisations, where novel techniques are employed for the first time (Altamimi et al, 2016 and 2023 respectively).

There is a lot of repetition in the text concerning the selection of stations on the basis of numbers of NPs, and Table 2 in particular displays far too many parameters for what is actually a simple regression result for each station velocity.

The paper could be saved by carrying out a strong edit and providing more detail on force models, tidal loading, etc., but I fear that not taking into account the recent progress on mitigation of systematics will make many of the conclusions difficult to sustain.

Some comments are given in the annotated copy of the manuscript.

Author Response

Response to Reviewer 1 Comments

Remote Sensing, manuscript ID: remotesensing-2459947

Title: Analysis of the results of determining position and velocity of the Satellite Laser Ranging stations in the time of the earthquakes 2010-2011
Authors: Stanisław Schillak *, Agnieszka Satarowska, Dominik Sankowski, Piotr Michałek

Thank you for review of the manuscript.

The paper sets out to discuss the geodetic use of laser ranging observations from the global network from a five-year period during which two major earthquakes took place. The paper's stated aim is to monitor the geocentric positions and velocities of two of the stations most affected by the earthquakes, as well as an assessment of the quality of the coordinates retrieved for all the stations used in the analysis.

Point 1: An overall impression of the paper is that not enough detail is given about the force model and station displacement models used by the GEODYN-II analysis package.

Response 1: All significant force models and parameters used by GEODYN-II software are presented in Table 1. The station displacement models are not determined by orbital analysis but on the base of the stations positions for arc reference epoch (first day of the month). The comparison with tectonic plates models e.g. NNR-NUVEL-1A or others, give very good agreement in plate direction and velocity.

Point 2: The paper also feels very much 'of of date' with work ignored on improved SLR analyses in recent years;

Response 2: "The aim of the work was to detect jumps and changes in the position and velocity of the SLR stations, not to eliminate them. This is very important information for SLR stations and analysis data centers. For this purpose, the period 2008-2012, interesting in the authors' opinion, was selected, in which there were two strong earthquakes, which additionally caused three stations to move several centimeters. On the other hand analysis centers, especially the ILRS Analysis Standing Committee have different task. They have to determine the station coordinates as accurate as possible. It is especially important for determination of International Terrestrial Reference Frame, or satellites coordinates, where jumps can not to exist". This paragraph was added in Introduction.

Point 3: for instance no reference is made to improved models for centre-of-mass corrections for the geodetic satellites (Rodriguez et al, 2019)

Response 3: Introducing the center-of-mass corrections depending on the SLR stations is a very important task, but for the period under review, these corrections were not known to all stations, and hence the decision not to introduce them for the sake of data homogeneity. On the other hand, these changes are systematic constants and are very small, so their absence cannot distort the final results.

Point 4: and the ILRS Analysis Standing Committee work on mitigation of known systematic errors in the stations (Luceri, et al, 2019), now adopted by all the Analysis Centres and used for the ILRS input to the new version of the reference frame (ITRF2020, Altamimi et al, 2023).

Response 4: Eliminating station systematic biases by entering range biases is a good idea, but it is often better to remove erroneous results if we do not know the cause of the errors.

Point 5: There is also a newer reference for the ILRS (Pearlman et al, 2019).

Response 5: An older and newer reference for ILRS is in line 44 [3, 4]. This sentence has been little corrected "Satellite Laser Ranging representing by International Laser Ranging Service (ILRS) [3, 4] as an absolute measurement technique, by means of the obtained data, determine the position of the axis of the Earth's geocentric reference system XYZ and the center of the system, which is the Earth's center of mass. "  

Point 6: To ignore these improvements in data handling means that many of the conclusions about station stability are not relevant and very misleading to the reader and users of the SLR technique.

Response 6: In this article, we want to present the results taking into account of all jumps and inaccuracies. This is especially important for many SLR stations that have been observing without information about systematic errors, which, when eliminated, can significantly improve the quality of the stations results. The differences between the ILRS solution and the results presented in this article are not too significant. We need to find an answer to the fundamental questions, why is the accuracy of SLR so much lower than GNSS and VLBI? Why do we have such significant differences between SLR stations? The analysis presented in the article allows for the assessment of individual SLR stations, their problems and possibilities.

Point 7: The motions of the two targeted stations as affected by the earthquakes are well-described and useful results from the technique. But no reference at all is made to the work presented in the ITRF 2014 and ITRF 2020 realisations, where novel techniques are employed for the first time (Altamimi et al, 2016 and 2023 respectively).

Response 7: Yes, in two new ITRF solutions, ITRF2014 and ITRF2020, were introduced Post Seismic Deformation (PSD) Model, and Annual and Semi-annual frequencies. The PSD does not apply to the stations presented in this work, both stations in Concepcion and Koganei were closed in 2014, only the Arequipa station could be used PSD, but due to the small number of NPs this station is not analyzed in this work. Due to the need of showing changes in the position and velocity of the station, ITRF2008 was used, which contains the coordinates before the quakes, then changes in the station coordinates as a result of the quake are much better illustrated. The differences between ITRF2008, ITRF2014 and ITRF2020 are not so great. ITRF2014 and ITRF2020 due to new coordinates after quake introduce significant smoothing of the studied effects, making it impossible to obtain a real picture of the effects of the quake.

Point 8: There is a lot of repetition in the text concerning the selection of stations on the basis of numbers of NPs,

Response 8: NPs play a very important role in the precision and accuracy of determining station coordinates, hence a lot of similar phrases in the text of the work.

Point 9: and Table 2 in particular displays far too many parameters for what is actually a simple regression result for each station velocity.

Response 9: Some columns in Table 2 have been canceled.

Point 10: The paper could be saved by carrying out a strong edit and providing more detail on force models, tidal loading, etc., but I fear that not taking into account the recent progress on mitigation of systematics will make many of the conclusions difficult to sustain.

Response 10: We believe that the presented results correspond to real changes in the position and velocity of the SLR stations and can be an example of assessing the quality of laser stations to remove the causes of measurement errors and eliminate the real changes in SLR station coordinates. The force models, tidal loading, etc. were introduced in the orbital computations. Presented in this paper analysis allows to evaluate individual SLR stations, their problems and possibilities.

Point 11: Some comments are given in the annotated copy of the manuscript.

Response 11: All corrections and comments were corrected in the text without line 259, 261,262 (LAGEOS), determination of range biases and orbital RMS is possible separately for both LAGEOS satellites, it is also good control of correctness of the computations.

Reviewer 2 Report

It is of great importance to accurately determine position and velocity of SLR stations for their supports to the realization of ITRFs. In this study, authors made use of measurements from SLR stations during the 2008-2012 period and determined position and velocity for each station with consideration of great earthquakes during this period. However, in my opinion, there are a couple of issues needed to be clarified and I suggest a major revision.

The major comments:

1) The novelty of this work is necessary to be strengthened. Authors adopted the routine method and software to process the measurements from SLR stations and finally determined position and velocity of the stations. During the process, authors claimed that the influence of great earthquakes were considered, but the detailed strategy to exclude co- and post-seismic effects induced by great earthquakes was not mentioned. Afterall, the removal of earthquake effects is critical in determining the position and velocity of the near-field SLR station, since inaccurate modeling of earthquake effects would bias the results. Thus, I suggest authors provide the detailed processing of earthquake effects to strengthen the novelty of this work.

2) The result about position and velocity of SLR stations is not convincing. Authors just solved station position and velocity by using the common procedure in SLR data processing, but they did not compare their results with that from GNSS or VLBI stations collocated. Since the collocated stations would undergo the same or similar earthquake effects and be with the same or similar velocities. Therefore, it’s necessary to do the comparative analysis between SLR-derived results and results from other space satellite technologies, such as GNSS or VLBI.

3) In section 5 “Discussion and Conclusions”, authors merely introduced the changes caused by the great earthquakes or other reasons, but they did not discuss their influences on the determination of station position and velocity. Similar to comment 2), the precision and accuracy of the solve results should be discussed in this part, in addition, the unique contribution of this work is also necessary to be strengthened in this part.

4) It’s very necessary to illustrate the locations of SLR stations by a figure, and the solved velocity field of SLR stations in another figure.

5) Some minor comments:

a. Line 97, “height above sea level”?

b. Line 121-122, “The position is ……” is not clear.

c. Line 138, what does “O-C” mean?

d. Line 217, what’s the reference to equation (10)?

A careful proofreading is very necessary for some typos, grammar errors, incomplete sentences, and sentence repetitions.

Author Response

Response to Reviewer 2 Comments

Remote Sensing, manuscript ID: remotesensing-2459947

Title: Analysis of the results of determining position and velocity of the Satellite Laser Ranging stations in the time of the earthquakes 2010-2011
Authors: Stanisław Schillak *, Agnieszka Satarowska, Dominik Sankowski, Piotr Michałek

Thank you for review of the manuscript.

Point 1: It is of great importance to accurately determine position and velocity of SLR stations for their supports to the realization of ITRFs. In this study, authors made use of measurements from SLR stations during the 2008-2012 period and determined position and velocity for each station with consideration of great earthquakes during this period. However, in my opinion, there are a couple of issues needed to be clarified and I suggest a major revision.

Response 1: We propose adding in Introduction the following paragraph:

"The aim of the work was to detect jumps and changes in the position and velocity of the SLR stations, not to eliminate them. These effects are very important information for SLR stations and SLR analysis data centers. For this purpose, the period 2008-2012, interesting in the authors' opinion, was selected, in which there were two strong earthquakes, which additionally caused three stations to move several centimeters. On the other hand analysis centers, especially the ILRS Analysis Standing Committee have different task. They have to determine the station coordinates as accurate as possible. It is especially important for determination of International Terrestrial Reference Frame, or satellites coordinates, where jumps should not occur."

The major comments:

Point 2: 1) The novelty of this work is necessary to be. Authors adopted the routine method and software to process the measurements from SLR stations and finally determined position and velocity of the stations. During the process, authors claimed that the influence of great earthquakes were considered, but the detailed strategy to exclude co- and post-seismic effects induced by great earthquakes was not mentioned. Afterall, the removal of earthquake effects is critical in determining the position and velocity of the near-field SLR station, since inaccurate modeling of earthquake effects would bias the results. Thus, I suggest authors provide the detailed processing of earthquake effects to strengthen the novelty of this work.

Response 2: Yes, we agree that the novelty of this work should be strengthened. We suggest adding to the text of this article in Section 5 Discussion and Conclusions the following paragraphs:

“In opinion of the authors, the most important contribution of this work is the demonstration of a square change in the position of the station after strong earthquakes, which is shown in down side of the Figures 11 and 12. This allows to determine the position and velocity of the station at arbitrary moment after the earthquake, and thus easier to remove the effects of strong earthquakes compared to the one used in ITRF2014 and ITRF2020 the post-seismic deformation model (PSD).

Another important information obtained in this work is the detected jumps of 2-3 cm for three SLR stations; Chanchun, Simosato (as an effect of the tsunami in Japan) and San Juan (as an effect of the Concepcion earthquake). These jumps can be easily eliminated by the ITRF in the form of additional station coordinates before and after the earthquake (this is implemented in the ITRF). The systematic errors found in this work for the three stations can be or have been eliminated by introducing appropriate range biases. Finally, the annual waves for the Haleakala station can be removed in ITRF2020 by entering Annual and Semi-annual Frequencies.

The most important thing is that the information about the observed changes in the location of the station should be monitored, for example in the form presented in this paper, and transmitted to the station as soon as possible. Such actions would improve the quality of the SLR stations, and thus bring the quality of SLR stations closer to the primary goal of the Global Geodetic Observing System (GGOS) of achieving a positioning accuracy of 1 mm and a velocity of 0.1 mm/year.”

Point 3: 2) The result about position and velocity of SLR stations is not convincing. Authors just solved station position and velocity by using the common procedure in SLR data processing, but they did not compare their results with that from GNSS or VLBI stations collocated. Since the collocated stations would undergo the same or similar earthquake effects and be with the same or similar velocities. Therefore, it’s necessary to do the comparative analysis between SLR-derived results and results from other space satellite technologies, such as GNSS or VLBI.

Response 3: The comparison position and velocity of SLR stations with similar results from GNSS needs additional work, but this task was performed by author in several previous papers. List of these works is presented below:

Schillak S., Lehmann M., 2009, The comparison of the station coordinates between SLR and GPS, in: Schillak S. (Ed.), Proc. 16th International Workshop on Laser Ranging, Poznań, 13-17.10.2008, pp. 176-182.

Szafranek K., Schillak S., Araszkiewicz A., Figurski M., Lehmann M., Lejba P., 2012, Comparison of long-term SLR and GNSS solutions from selected stations in the frame of GGOS realization, Geophysical Research Abstracts, Vol. 14, EGU2012-6087, EGU General Assembly 2012, 22-27 April, Vienna, Austria.

Schillak S., Luck J., Szafranek K., Lehmann M., Lejba P., 2012, The comparison of the local geodetic tie and the results of orbital analysis between SLR and GNSS stations in Orroral and Mt. Stromlo, Abstracts 39th COSPAR Scientific Assembly, ISSN-1815-2619, PSD.1-0034-12, 2012, 14-22 July, Mysore, India

Szafranek K., Schillak S., 2012, Introduction to joint analysis of SLR and GNSS data, Reports on Geodesy, vol. 92, no. 1, pp. 143-154, 2012.

Szafranek K., Schillak S., Araszkiewicz A., Figurski M., Lehmann M., 2012, Results of analysis of long-term SLR and GPS solutions made for geophysical purposes, AGU Abstracts G53B-1141, AGU Fall Meeting 2012, 3-7 December 2012, San Francisco.

Szafranek K., Schillak S., Araszkiewicz A., Figurski M., Lehmann M., Lejba P., 2013, Local ties verification based on analysis of long-term SLR and GPS solutions, Geophysical Research Abstracts, Vol. 15, EGU2013-4548-1, EGU General Assembly 2013, 07-12 April, Vienna, Austria.

Schillak, S.; Szafranek, K.; Araszkiewicz, A.; Figurski, M.; Lejba, P.; Lehmann, M. Earthquakes problem in ITRF position and velocity determination from SLR and GNSS data. IAG Scientific Assembly, Potsdam, Germany, 1 September 2013.

Bogusz J., Figurski M., KÅ‚os A., Schillak S., Szafranek K., 2014,  Long-term dependencies on selected GPS-SLR co-located sites, Geophysical Research Abstracts, Vol. 16, EGU2014-1596, EGU General Assembly 2014, 27 April – 2 May, Vienna, Austria.

Sapota M., Szafranek K., Bogusz J., Figurski M., Schillak S., Nykiel G., 2014, Determination of post-seismic decays from selected GNSS and SLR co-located sites, 14th SGEM GeoConference on Informatics, Geoinformatics and Remote Sensing, 19-25 June 2014, SGEM2014 Conference Proceedings, Vol. 2, DOI: 10.5593/SGEM2014/B22/S9.026, ISBN 978-619-7105-11-7/ISSN 1314-2704, pp. 199-206.

We chose two papers as reference in this article, first present station positions and velocities from SLR and GNSS results for the earthquake time in 2010-2011:

Schillak, S.; Szafranek, K.; Araszkiewicz, A.; Figurski, M.; Lejba, P.; Lehmann, M. Earthquakes problem in ITRF position and velocity determination from SLR and GNSS data. IAG Scientific Assembly, Potsdam, Germany, 1 September 2013

and second for all SLR-GNSS collocation sites:

Szafranek K., Schillak S., Araszkiewicz A., Figurski M., Lehmann M., Lejba P., 2012, Comparison of long-term SLR and GNSS solutions from selected stations in the frame of GGOS realization, Geophysical Research Abstracts, Vol. 14, EGU2012-6087, EGU General Assembly 2012, 22-27 April, Vienna, Austria.

We detect very good agreement for the results from both techniques. In the all papers we used the same method of the SLR station position and velocity determination as in this article. For VLBI is too small number of collocation sites for more detailed analysis.

Point 4: 3) In section 5 “Discussion and Conclusions”, authors merely introduced the changes caused by the great earthquakes or other reasons, but they did not discuss their influences on the determination of station position and velocity. Similar to comment 2), the precision and accuracy of the solve results should be discussed in this part, in addition, the unique contribution of this work is also necessary to be strengthened in this part.

Response 4: All detected changes in the station positions and velocities and the unique contribution of this work are discussed in section 5 in additional part presented in comment 1). The detailed influences on the station position and velocity are presented in Appendix in Tables A2 (positions) and A3 (velocities). More detailed analysis of these inaccuracies is presented in section 5.

Point 5: 4) It’s very necessary to illustrate the locations of SLR stations by a figure, and the solved velocity field of SLR stations in another figure.

Response 5: Sorry, we don't understand these proposed figures. Locations of all SLR stations are shown in Appendix in Table A1, velocities 3D, 2D, vertical and azimuth of the station motion in Table A3.

5) Some minor comments:

  1. Line 97, “height above sea level”?

should be "geodetic height" - corrected in the text

  1. Line 121-122, “The position is ……” is not clear.

should be "The station position is computed by GEODYN-II software from the laser range measurements for monthly arc for a common reference epoch (2005.0) (detrended data)." - corrected in the text

  1. Line 138, what does “O-C” mean?

should be "gives the difference observed-computed values (O - C) (Δρ):" -  corrected in the text

  1. Line 217, what’s the reference to equation (10)?

should be "The total RMS for all components (total station position stability) in reference to ITRF2008 was computed from the formula (10):" - corrected in the text

Comments on the Quality of English Language

A careful proofreading is very necessary for some typos, grammar errors, incomplete sentences, and sentence repetitions.

English language was improved.

Reviewer 3 Report

I have enjoyed reading your manuscript; a couple of comments, that may open the way to further and future improvements:

1. As you correctly stated in the conclusions, it would be nice to see the work extended to include other satellites:

a. How better (or worse) do the results get adding (or recomputing from scratch with) the Etalon’s, and/or the LARES’s, and/or etc., to the analysis?

b. How do the results change mixing up multiple (at least two) families of cannon-ball satellites (LAGEOS w/ Etalon, LAGEOS w/ LARES, Etalon w/ LARES, etc.)?

2. The only ambiguity that I see in your work may concern the actual engineering structures of the buildings whose positions and velocities changed: were those (the structures) alike and comparable in terms of quake-proof capabilities?

Even if I am not qualified to assess the quality of English in this paper, because I am not a native English speaker, I have noticed a few typos: perform a deeper proof-read.

Author Response

Response to Reviewer 3 Comments

Remote Sensing, manuscript ID: remotesensing-2459947

Title: Analysis of the results of determining position and velocity of the Satellite Laser Ranging stations in the time of the earthquakes 2010-2011
Authors: Stanisław Schillak *, Agnieszka Satarowska, Dominik Sankowski, Piotr Michałek

I have enjoyed reading your manuscript; a couple of comments, that may open the way to further and future improvements:

Thank you for review of the manuscript and your nice opinion.

Point 1: 1. As you correctly stated in the conclusions, it would be nice to see the work extended to include other satellites:

  1. How better (or worse) do the results get adding (or recomputing from scratch with) the Etalon’s, and/or the LARES’s, and/or etc., to the analysis?

Response 1: The International Laser Ranging Service (ILRS) uses in the process of determination of the International Terrestrial Reference Frame (ITRF) and other analysis four satellites: LAGEOS-1, LAGEOS-2, Etalon-1 and Etalon-2. Due to the small number of observations of the Etalon satellites, their impact on the final result is imperceptible. Therefore, most SLR analysis centers use to determine station coordinates only the LAGEOS-1 and LAGEOS-2 satellites. Attempts to include the LARES-1 satellite were made by the ILRS, as well as by the authors of this paper (Schillak et al., 2021). The results in common solution with the LAGEOS satellites were promising. Currently, we have the LARES-2 satellite, which is similar in parameters to the LAGEOS and will certainly be introduced in future as the third basic satellite for station coordinates determination. It takes some time to verify the results from this satellite. The authors intend to prepare a publication on this subject in the near future. Other low geodetic satellites were tested for station coordinates determination, but the results were unsatisfactory.

Schillak, S.; Lejba, P.; Michałek, P., 2021, Analysis of the Quality of SLR Station Coordinates Determined from Laser Ranging to the LARES Satellite. Sensors, 21(3), 737. https://doi.org/10.3390/s21030737

Point 2: b. How do the results change mixing up multiple (at least two) families of cannon-ball satellites (LAGEOS w/ Etalon, LAGEOS w/ LARES, Etalon w/ LARES, etc.)?

Response 2: Mixing LAGEOS satellites with Etalons does not improve the results of determining station coordinates. LAGEOS with LARES-2 raises hopes for an improvement in results. The use of the multi-satellite method, i.e. all geodetic satellites combined, is being considered, but no results have been obtained so far.

Point 3: 2. The only ambiguity that I see in your work may concern the actual engineering structures of the buildings whose positions and velocities changed: were those (the structures) alike and comparable in terms of quake-proof capabilities?

Response 3: Strong earthquakes in 2010-2011 affect two stations: Concepcion (Chile) and Koganei (Japan). A very strong quake in Concepcion caused significant damage to a small mobile SLR pavilion and toppled a laser telescope. A slightly weaker quake in Japan as a result of the tsunami did not cause any damage to the SLR pavilion due to its massive construction and considerable distance from the epicenter (I had the opportunity to visit both stations before the quake).

Comments on the Quality of English Language

Even if I am not qualified to assess the quality of English in this paper, because I am not a native English speaker, I have noticed a few typos: perform a deeper proof-read.

English language was improved.

Round 2

Reviewer 1 Report

I still find the results to be somewhat muddled, with the determination of station systematic errors mixed in with the relatively more interesting post-earthquake motions of a subset of the network.

As previously reported by this reviewer regarding the first draft of the paper, the ILRS has come up with a procedure that shows that systematic errors can be solved for simultaneously with the station coordinates, so that real station motions can be determined. It is not true to say (line 55), that 'The vertical changes (U) resulting mainly from systematic errors of the 55 SLR stations.'

I would prefer to see a clear distinction in the results of this work that separates 'station systematic errors' from 'station motion due to earthquakes'. Headed sub-sections in the 'Discussions and Conclusions' part of the work would clarify this distinction.

Author Response

Response to Reviewer 1 Comments 2

Remote Sensing, manuscript ID: remotesensing-2459947

Title: Analysis of the results of determining position and velocity of the Satellite Laser Ranging stations in the time of the earthquakes 2010-2011
Authors: Stanisław Schillak *, Agnieszka Satarowska, Dominik Sankowski, Piotr Michałek

Point 1: I still find the results to be somewhat muddled, with the determination of station systematic errors mixed in with the relatively more interesting post-earthquake motions of a subset of the network.

Response 1: Systematic errors are a very important parameter of any measurement and their evaluation is often a key measurement task. Our task is to determine the station coordinates and change them over time. Based on the analysis of the results, it is difficult to distinguish whether the shift of the station is real or caused by systematic instrumental errors (apparent shift). Can we treat real systematic shifts as errors - no, as deviations - yes. Each of the types of shifts is presented in the paper in separate subsections: earthquake effects in subsection 4.2.1. "Earthquakes", instrumental systematic errors in subsection 4.2.2. "Other systematic shifts".

Point 2: As previously reported by this reviewer regarding the first draft of the paper, the ILRS has come up with a procedure that shows that systematic errors can be solved for simultaneously with the station coordinates, so that real station motions can be determined.

Response 2: Yes, in the process of computing orbits, determining station coordinates and additional procedures such as Post Seismic Deformation Model (PSD), and Annual and Semi-annual frequencies, most of the systematic errors can be eliminated, for the most accurate few stations it is fulfilled, as shown by the results of this work (6 stations have no comments regarding their quality problems). The remaining 15 analyzed stations show significant shifts in position caused by real or apparent changes. The deterioration of the quality of these stations is also caused by the increase in the random error of determining the coordinates. Knowing all these changes is very important for improving the quality of SLR stations.

Point 3: It is not true to say (line 55), that 'The vertical changes (U) resulting mainly from systematic errors of the SLR stations.'

Response 3: I do not agree with this opinion. Vertical changes, with the exception of SLR systematic errors, are caused by the following reasons (the motion of tectonic plates does not cause changes in the vertical component):- by strong earthquakes (as shown in this work, three stations in total)- by local shifts (very rarely),- by volcanic effects (only one such case was found - Tateyama station in Japan (Schillak S., Wnuk E., Kunimori H., Yoshino T., 2006, Crustal deformation in the Key Stone network detected by satellite laser ranging, Journal of Geodesy, Vol. 79, (12), pp.682-688),- by post-glacial uplift (does not include SLR stations so far),- by tropospheric delay (very small effect),- by correction for the center of mass of the satellite (very small effect).All in all, there are very few such vertical shifts. Other vertical effects are computed in the process of determining of satellite orbits.I think that more than 90% of vertical changes are the result of instrumentation errors, i.e. lack of proper level for registering the "start" and "stop" pulses, improper adjustment of the calibration system, systematic error in determining the distance to the target, systematic error in determining the intersection of the telescope axes or the marker (reference point of the SLR station), incorrect level of sensitivity for the reflected pulses, time registration error, etc. (Schillak S., 2004, Analysis of the process of the determination of station coordinates by the satellite laser ranging based on results of the Borowiec SLR station in 1993.5-2000.5, Part 1; Performance of the satellite laser ranging, Artificial satellites, Vol. 39, No. 3, Warsaw, Poland, pp. 217-263).

Point 4: I would prefer to see a clear distinction in the results of this work that separates 'station systematic errors' from 'station motion due to earthquakes'. Headed sub-sections in the 'Discussions and Conclusions' part of the work would clarify this distinction.

Response 4: We cannot distinguish from changes in station coordinates alone that they were caused by earthquakes or systematic instrumental errors. Only additional information about the location of the earthquake and comparison with the GNSS results allows to distinguish what these deviations are caused by. In order to improve the transparency of the work, 5 subsections were introduced in the section "Discussions and Conclusions", which better illustrate the obtained results: 5.1. Changes of position caused by earthquakes, 5.2. Changes in position caused by systematic errors, 5.3. Station velocities, 5.4. Accuracy, 5.5. Conclusions.

Thank you for your comments.